# Effect of Sulfadimethoxine, Oxytetracycline, and Streptomycin Antibiotics in Three Types of Crop Plants—Root, Leafy, and Fruit

**Reep Pandi Tasho [1], Song-Hee Ryu [2] and Jae-Young Cho [3],***

1   Department of Agriculture Chemistry, Chonbuk National University, Building 3-2, Room 116, Jeonju-si 56756, Jeollabuk-do, Korea; reeplepcha@gmail.com
2   Chemical Safety Division, Agro-Food Safety and Crop Protection Department, National Institute of Agricultural Sciences, Rural Development Administration, Jeonju-si, 55365, Korea; tendergreen@korea.kr
3   Department of Agriculture Chemistry, Chonbuk National University, Building 3-2, Room 106, Jeonju-si 56756, Jeollabuk-do, Korea
*   Correspondence: soilcosmos@jbnu.ac.kr

**Abstract:** (1) Background: Plants act as the natural sink for a variety of toxins in the environment, including *veterinary antibiotics* (VAs). The objective of this study was to evaluate the uptake and fate of *sulfadimethoxine* (SDZ), *oxytetracycline* (OTC), and *streptomycin* (STR) in lettuce (*Lactuca sativa* L.), carrot (*Daucus carota*), and pepper (*Capsicum annum*) grown in VAs amended soil. (2) Methods: 0, 50, and 100 mg kg$^{-1}$ VA laced manure was applied in a sandy clay loam soil. (3) Results: 30-d (lettuce) and 60-d (carrot and pepper) greenhouse experiment showed that SDZ and OTC were taken up by all three plants, with concentrations in plant tissue ranging from 0.1 to 1.2 mg kg$^{-1}$ dry weight. The concentration of VAs in plant tissues increased with a corresponding increase of antibiotics in manure. The highest plant tissue concentrations were found in carrot and lettuce, followed by pepper. An increase in NADPH P450 reductase and glutathione-s-transferase enzyme activity with increasing SDZ and OTC concentration was evident, signifying the induction of the detoxification process. The activity of plant detoxification enzymes under STR treatment was found not to be significantly different from control. (4) Conclusions: These results raise potential human health concerns of consuming low levels of antibiotics from produce grown on manure-amended soils. The result indicates that SDZ, OTC, and STR antibiotics posed high, medium, and low acute ecological risks in lettuce, carrot, and pepper plants when grown in sandy clay loam soil.

**Keywords:** carrot; lettuce; pepper; veterinary antibiotics; plants; soil

## 1. Introduction

Antimicrobials are widely used in veterinary medicine and can contaminate the environment. The contamination of soil with such antimicrobials, especially agricultural fields receiving frequent manure, is a universal phenomenon. Once in the environment, they may sorb onto soil and sediment, be transported to ground or surface waters, or be degraded. Even though different types of antibiotics have different anticipated environmental exposure routes, the application of animal manure containing excreted antibiotics remains the predominant pathway for veterinary antibiotics (VAs) release into the terrestrial environment [1,2]. Once in the arable land, VAs can impact soil microbial activity as well as vegetation growth [3]. Plants act as the natural sink for a variety of toxins in the environment, including VAs. Once absorbed into the plant system VAs, depending upon their property, get transformed, conjugated, and then finally sequestrated. Since plants are unable to excrete the toxins and their

metabolites, they get sequestrated into vacuoles or cell walls. Therefore, there is a possibility for the transfer of VAs and their metabolites through the consumption of contaminated plants.

The uptake of VAs by plants is a broad and growing area of research, and several studies have been conducted previously [4–6]. However, unlike pesticides and herbicides, antibiotics as potential pollutants have not aroused attention until reasonably recently [7]. In addition, the relevance of antibiotic ecotoxicology is scarce due to the minimal possible effects of antibiotics in the environment [8]. Phytotoxicity tests of VAs using seed germination and plant growth test have been conducted previously [9,10]. Studies have shown the effect of VAs on plants to be different between antibiotic compounds and between plant species [3,11,12]. Therefore, studying the environmental implication of VAs associated with animal manure application by considering plant species variability is of immense importance.

This study investigates the effect of *oxytetracycline* (OTC), *sulfadimethoxine* (SDZ), and *streptomycin* (STR) on three types of plants—root (carrot), leafy (lettuce), and fruit (pepper). The purpose of this study was (1) to determine the uptake and accumulation of VAs by lettuce, carrot, and pepper, (2) to determine the time-dependent effects of the VAs on mycorrhizal frequency, proline content, and detoxifying enzymes, (3) to assess environmental risk assessment of the VAs.

## 2. Materials and Methods

### 2.1. Plant

Three different types of crop plants with different characteristics were used

- Leafy: Lettuce, *Lactuca sativa*, Asteraceae.
- Root: Carrot, *Daucus carota*, Apiaceae.
- Fruit: Pepper, *Capsicum annum*, Solanaceae.

### 2.2. Veterinary Antibiotics

Three VAs (OTC, SDZ, and STR) covering a wide range of molecular weight were purchased from Sigma Aldrich Korea. Table A1 presents the structure and properties of the chosen antibiotics.

### 2.3. Phytotoxic Assessment

The filter method, according to the International Seed Testing Association [13], was carried out. Plant seeds (5 seeds per petri dish) were placed in Petri dishes (10 cm) containing 9 different concentrations (0, 1, 5, 10, 50, 100, 300, 500, and 1000 mg kg$^{-1}$) of VAs soaked (5 mL) filter paper (9 cm). Seeds were germinated in an incubator at 25 °C in darkness. The test was conducted in triplets and repeated three times. The root length of seedlings (primary root ≥ 5 mm) was the endpoint for evaluation, which amounted to 6 days for this study [14]. The determination of the EC$_{50}$ germination index (%) and root growth percentage calculated at the end of 6 days was done according to Tiquia et al. 1996 [15].

The effect of the three selected VAs on plant growth was assessed in sandy clay loam soil (obtained from a rural site in Chonbuk National University). Table 1 presents the physicochemical property of the soil. Inorganic fertilizer added at the rate of 1.5 kg per 50 kg soil to meet the adequate nutrient requirement. The final modified soil comprised of pH 5.59 and total N, P, K values of 3257.8, 455.6, and 3647 mg kg$^{-1}$, respectively.

VAs were firstly mixed with animal manure (pH 6.4, total organic matter: 53.22%) and then introduced into the modified soil to make up the total organic matter content of 2.5%. The test was conducted with nine concentrations of the VAs (0, 1, 5, 10, 50, 100, 300, 500, and 1000 mg kg$^{-1}$) in triplicates. Seeds were sowed into 4 kg soil at a depth of 1 cm in a pot (15 × 15 × 15 cm), watered (60% moisture content), weighed, and placed in an open greenhouse. Plants were watered 4–5 times a week

with regular tap water. Measurement of root and shoot length after 15 days, for the determination of $EC_{50}$ values.

**Table 1.** Properties of the soil used in the study.

| Soil Type | Sand (%) | Silt (%) | Clay (%) | OM (%) | pH (DW) | EC (μS/cm) | N (mg kg$^{-1}$) | P (mg kg$^{-1}$) | K (mg kg$^{-1}$) |
|---|---|---|---|---|---|---|---|---|---|
| Sandy clay loam | 59.47 | 9.8 | 30.73 | 1.23 | 5.23 | 72.85 | 518 | 379.1 | 3400 |

Based on the $EC_{50}$ values from the plant growth test, three different test concentrations of the VAs were selected for the study (10, 50, and 100 mg kg$^{-1}$). In addition, sandy clay soil with no added VAs (0 mg kg$^{-1}$) was used as a control for the study. Lettuce, carrot, and pepper seeds were then sown in the control and test soil following the same procedure as mentioned above. Lettuce plants were harvested after 30 days, whereas carrots and pepper after 60 days—the plant samples were then used for the assessment of VAs toxicity in plants.

### 2.4. Veterinary Antibiotics Accumulation and Bio-Concentration

At the end of the growth period, plants were harvested carefully, and different plant parts analyzed for VAs.

- Lettuce: Root
- The bottom part of the leaf
- The apical part of the leaf
- Carrot: Leaves

  - Foot (base): peel, flesh
  - Brain (upper): peel, flesh

- Pepper: Root
- Stem
- Leaves
- Fruit

The designated plant parts chopped separately, homogenized with liquid $N_2$ and stored at −20 °C until analysis. VAs in plant root and soil, after 49 days, were extracted using the QuECHers AOAC LC-MS sample preparation kit. VAs in run-off water, during watering, was not calculated as the purpose of this study does not call for it. Briefly, 1 g sample (in 8 mL DW) extracted with a 12 mL QuECHers buffer solution. Following ultrasonication (15 min), QuECHers buffering salt was added and centrifuged for 15 m, 1500 rpm. The pH of the supernatant was made to 7.5 and centrifuged for 3 m, 3000 rpm. Then, 2 mL supernatant was transferred into a d-SPE cleanup tube, centrifuged (15 m, 15,000 rpm) and filtered through 0.2 μM filter. HPLC conducted in positive electron ionization mode with mobile phase A: 0.1% formic acid in DW and mobile phase B: 0.1% formic acid in acetonitrile kinetex 2.6u C18 50 × 2.1 mm at the flow rate of 0.5 mL min$^{-1}$. The gradient program was set as −0 min, A:B = 95:5; 1 min, A:B = 95:5; 10 min, A:B = 0:100; 12 min, A:B = 0:100; 15 min, A:B = 95:5; and 20 min, A:B = 95:5. VAs concentration was analyzed with Agilent 1200 series HPLC interfaced with Agilent G1314C variable wavelength detector, G1312B binary pump, and G1316B column SL.

Mass spectrometry analysis was conducted in positive electron ionization mode with the mobile phase flow into the chamber set at 11 mL min$^{-1}$, and the source temperature held at 300 °C. Detector signal for amino acids was recorded from 0–21 min and ions scanned across 190–400 *m/z*.

*Bioconcentration factor* (BCF) is the estimate used to determine the absorption potential of VAs by plants. It is the factor used for describing the rate of uptake of antibiotics from the soil by plants. BCF was calculated using the formula described by Zayed et al. [15].

$$BCF\ (\%) = \frac{Concentration\ root + Concentration\ aerial\ part}{Initial\ concentration\ in\ external\ medium}$$

### 2.5. Detoxification of Veterinary Antibiotics

According to the "green liver" concept, that describes the fate of organic contaminants in plants, the metabolism of xenobiotics in plants also serves the detoxification purposes [16]. Metabolism of VAs in plants usually occurs in three phases—transformation (Phase I), conjugation (Phase II), and compartmentalization (Phase III). Additionally, detoxifying enzymes play essential roles in the control of inflammation, cellular redox status, and cytotoxicity [17]. In this study, only Phase I and Phase II enzymes involved in the metabolism of endogenous reactive compounds, as well as xenobiotics, including VAs, were studied. Phase I *NADPH-cytochrome P450 reductase* (CPR) reductase, and Phase II *Glutathione-s-transferase* (GST) enzyme activity was assessed. CPR transfers electrons from NADPH to cytochrome P450 and catalyzes the one-electron reduction of many drugs and foreign compounds. CPR activity was determined spectrophotometrically using the cytochrome c Reductase (NADPH) Assay Kit (CY0100) from Sigma Aldrich. The determination was based on a colorimetric assay that measures the reduction of cytochrome c by NADPH-cytochrome c reductase in the presence of NADPH. The reduction of cytochrome c results in the formation of distinct bands in the absorption spectrum and the increase in absorbance at 550 nm. On the other hand, the methodology described by Habig et al. was used for the GST conjugation reaction, where the ability of GST enzyme to conjugate 1-chloro-2, 4-dinitrobenzene and glutathione resulting in the change in absorbance at 340 nm [18].

### 2.6. Secondary Experiments

Mycorrhizal frequency and free proline content of lettuce, carrot, and pepper plants were determined. The intensity of mycorrhizal colonization calculated according to Trouvelot et al. [19]. Mycorrhizae were fixed and stained by incubating the roots ($\leq 2$ mm in diameter cut into 3 cm) in 10 mL 10% (*w/v*) KOH for 20 min (water bath, 80 °C). After incubation, roots were allowed to stand for 10 min at room temperature after the addition of 40 μL 30% H2O2 in 10 mL and KOH. Roots were then rinsed with tap water three times before incubating for 5 min at room temperature in 10 mL HCl (10% *v/v*). The staining agent used was 0.05% aniline blue in 85% (*w/w*) lactic acid (water bath: 80 °C, 30 min) [20]. After mounting, the root tips were observed under a compound microscope at 20× and 40× magnification.

For determination of free proline, 0.5 g plant sample was homogenized in 10 mL of 3% aqueous sulfosalicylic acid. After filtration (Whatman's No. 2), 2 mL filtrate was mixed with 2 mL acid-ninhydrin and 2 mL of glacial acetic acid in a test tube. The mixture was placed in a water bath for 1 h at 100 °C. The reaction mixture extracted with 4 mL toluene and the chromophore containing toluene aspirated, cooled to room temperature, and absorbance was measured at 520 nm—free proline concentration calculated according to Bates 1973 [21].

$$Free\ proline\ (\mu g\ g^{-1}\ FW) = \frac{\frac{(\mu g\ proline\ mL^{-1} \times mL\ toluene)}{115.5\ \mu g\ \mu mol^{-1}}}{\frac{sample\ (g)}{5}}$$

### 2.7. Ecological Risk Assessment

Risk quotient (RQ) is a useful tool to characterize the potential ecological risk of many contaminants in the environment. RQ of the three VAs to lettuce, carrot, and pepper was measured in sandy loam soil by following the EMA guideline [22]. RQ was calculated as the ratio between predicted environmental

concentrations (PEC) and predicted no-effect concentration (PNEC). PEC was estimated according to the European Union System for the Evaluation of Substances (EUSES) guidelines and software. CAS number, MW, melting point, boiling point, octanol-water coefficient, water-solubility, use pattern, emission data, and vapor pressure were the factors taken into consideration for PEC determination. PNEC was estimated by dividing the lowest $EC_{50}$, from plant growth test in soil, with a default assessment factor of 2.

## 3. Results and Discussions

### 3.1. Phytotoxic Assessment

Instead of the number of germinated seeds, root length was taken as an endpoint for analysis, an approach consistent with the previous study for organic contaminants [12,23]. Table 2 lists the $EC_{50}$ values of the VAs tested on lettuce, carrot, and pepper seeds after the seed germination test. Carrot seeds were found to be the most susceptible to VAs and had the lowest $EC_{50}$ value of 7.49 mg kg$^{-1}$ for SDZ antibiotics. On the other hand, pepper seeds presented the least susceptibility to the VAs. STR antibiotics were the least toxic compound, especially towards pepper seeds with $EC_{50}$ value > 300 mg kg$^{-1}$ (306.84 mg kg$^{-1}$). Comparatively, SDZ, and OTC antibiotics were more toxic with $EC_{50}$ values < 100 mg kg$^{-1}$.

**Table 2.** Toxicity data from seed germination tests for lettuce, carrot, and pepper (filter paper test).

| Veterinary Antibiotics | $EC_{50}$ (mg kg$^{-1}$) | | |
|---|---|---|---|
| | Lettuce | Carrot | Pepper |
| Sulfadimethoxine | 10.89 | 7.49 | 11.76 |
| Oxytetracycline | 13.15 | 9.11 | 28.06 |
| Streptomycin | 60.24 | 36.89 | 306.84 |

From the seed germination test, it was evident that the effect of VAs on plant seed germination varied depending on both the type of VAs and the plant species used. Among the three-plant species studied, carrot was the most susceptible to the VAs depending upon the antibiotic type. SDZ antibiotics were most toxic to plant seed germination whereas; STR antibiotic was the least toxic.

Based on the plant growth tests in soil, only SDZ antibiotics affected the growth of plants significantly, $EC_{50}$ value less than 100 mg kg$^{-1}$ (Table 3). SDZ was the most toxic and visible plant growth observed at concentrations of 300 mg kg$^{-1}$ and above. Similar observations have been seen by Migliore et al. 1998, where sulphamethoxine at 300 mg kg$^{-1}$ depressed the growth of *Hordeum disthicum* both in vitro and in soil [24]. In contrast to the seed germination test, which showed significant inhibitory effects, the $EC_{50}$ values of OTC were near or more than 200 mg kg$^{-1}$ for plant growth in soil. STR antibiotics showed no evident growth inhibition to the test plants.

**Table 3.** Toxicity data from plant growth tests in soil for lettuce, carrot, and pepper plants.

| Veterinary Antibiotics | Endpoint | $EC_{50}$ (mg kg$^{-1}$) | | |
|---|---|---|---|---|
| | | Lettuce | Carrot | Pepper |
| Oxytetracycline | Plant height | 167.103 | 155.608 | 325.307 |
| | Root length | 97.825 | 61.662 | 113.06 |
| Sulfadimethoxine | Plant height | 55.201 | 40.835 | 152.142 |
| | Root length | 14.983 | 16.919 | 47.7 |
| Streptomycin | Plant height | 484.38 | 306.995 | 893.606 |
| | Root length | 197.44 | 153.72 | 377.77p |

Through this study, we established species variability to VAs toxicity. As found through the seed germination test, pepper was the least sensitive to the VAs, which was translated in the plant growth tests in the soil as well (Table 3). On the other hand, carrot plants were the most sensitive to VAs administration. The negative effect of VAs in lettuce plants, sampled after 30 days of growth in soil, was evident. The increasing concentration of SDZ in the soil-manure mixture hampered both root length and foliar development. On the other hand, in comparison with control, root length, instead of foliar development, was more compromised with OTC and STR treatments. Previous studies have reported species variability to VAs treatment [5,12]. Chlortetracycline and oxytetracycline enhanced the growth of radish and wheat, but corn plants remained unaffected [5]. A study reported a higher sensitivity of sweet oat and rice in comparison with cucumber [12]. Based on the $EC_{50}$ values (in this study), only SDZ antibiotics may affect the growth of lettuce, carrot, and pepper plants in soil.

### 3.2. VAs Accumulation and Bio-Concentration

In this study, only the active compounds of SDZ and OTC were extracted from the plant samples (Figure 1). However, STR antibiotics could be successfully extracted from soil (Table 4), indicating the failure of test plants to absorb and translocate the antibiotic. Interestingly all three test plants absorbed SDZ and OTC but not STR antibiotics, which can be explained based on the differences in adsorption coefficients ($K_d$) of these antibiotics in soil. STR is a strongly basic antibiotic known to form strong complexes with the soil components, especially clay, making it highly unavailable [25]. Since the soil used in this study was sandy clay loam with 30.73% clay content, STR can quickly form strong complexes with clay. Another explanation is the higher molecular weight of STR antibiotics (728.69 g $mol^{-1}$), almost double the mass of SDZ and OTC molecule. Such a large molecule can't be taken up by plants easily, both in mass flow (transpiration) or as active uptake. Similar conclusions have been drawn from a previous study where tylosin antibiotic with a large molar mass of 916.1 g $mol^{-1}$ was not taken up by green onion, cabbage, and corn plants [6].

**Table 4.** Concentration of veterinary antibiotics extracted from the soil planted with lettuce, carrot, and pepper plants at the end of the experimental period.

| VAs (mg $kg^{-1}$) | Lettuce (mg $kg^{-1}$) | Carrot (mg $kg^{-1}$) | Pepper (mg $kg^{-1}$) |
|---|---|---|---|
| **Control** | 0.00 ± 0.00 | 0.00 ± 0.00 | 0.00 ± 0.00 |
| **SDZ 10** | 1.19 ± 0.829 [d] | 1.41 ± 0.069 [cd] | 1.97 ± 0.001 [d] |
| **SDZ 50** | 3.31 ± 1.797 [c] | 3.84 ± 2.214 [b] | 4.41 ± 0.057 [b] |
| **SDZ 100** | 12.47 ± 3.173 [a] | 14.34 ± 1.681 [a] | 16.88 ± 0.047 [a] |
| **OTC 10** | 0.43 ± 0.649 [e] | 0.39 ± 0.049 [e] | 0.23 ± 0.058 [e] |
| **OTC 50** | 1.19 ± 0.741 [d] | 2.03 ± 0.237 [c] | 1.72 ± 0.784 [d] |
| **OTC 100** | 4.95 ± 0.794 [b] | 3.84 ± 0.343 [b] | 3.51 ± 0.402 [c] |
| **STR 10** | 0.07 ± 0.049 [f] | 0.08 ± 0.784 [f] | 0.09 ± 0.851 [f] |
| **STR 50** | 0.41 ± 0.649 [e] | 0.33 ± 0.049 [e] | 0.41 ± 0.058 [e] |
| **STR 100** | 1.29 ± 2.262 [d] | 1.17 ± 4.275 [d] | 1.57 ± 3.627 [d] |

± represents standard error. Different letters in a column (a, b, c, d, e, f) are significantly ($p < 0.05$) different according to Duncan's multiple range test. SDZ = sulfadimethoxine; STR = streptomycin; OTC = oxytetracycline.

On the contrary, all three plants (lettuce, carrot, and pepper) absorbed SDZ and OTC antibiotics (Figure 1). The active molecules of SDZ and OTC were recovered from all the plant parts examined. The amount of SDZ and OTC extracted, from the plant parts, increased with increasing concentration of the VAs in the soil-manure mixture, indicating a higher absorption of VAs by plants as the amount of VAs increases in the soil-manure mix. The maximum amount of SDZ and OTC was extracted from carrot plants, followed by lettuce and pepper. The highest plant-tissue concentration was found in the carrot and lettuce plant, followed by pepper (Figure 1). The concentration of SDZ extracted from lettuce, carrot, and pepper tissues were ≈ 0.2%, 0.4%, and 0.1% of the amount applied to the soil through manure. On the other hand, OTC concentration in lettuce, carrot, and pepper tissues was ≈

0.1%, 0.2%, and 0.05%, respectively. A higher accumulation of VAs in the roots than in aerial parts was evident. In carrot roots, the concentration of VAs was higher in the peel/skin in comparison to the flesh. A similar observation where *Solanum tuberosum* tubers accumulated a higher percentage of sulfamethazine in their peels [26].

The calculated BCF values for root and shoots of lettuce, carrot, and pepper plants grown in VAs amended soil presented in Table 5. BCF values increased with increasing concentration of VAs in the soil-manure mixture. Among the three test plants, the highest BCF was reported in carrot plants, followed by lettuce and pepper. The BCF values for lettuce plants ranged from $0.065 \pm 0.001$ to $2.307 \pm 0.008$ and $0.227 \pm 0.004$ to $6.103 \pm 0.005$ µg kg$^{-1}$ for OTC and SDZ, respectively. In carrot, BCF for OTC and SDZ ranged from $0.010 \pm 0.054$ to $2.132 \pm 0.003$ and $0.018 \pm 0.023$ to $8.859 \pm 0.002$ µg kg$^{-1}$, respectively. On the other hand, BCF in pepper ranged from $0.033 \pm 0.010$ to $0.474 \pm 0.118$, and $0.066 \pm 0.005$ to $3.361 \pm 0.099$ µg kg$^{-1}$ for OTC and SDZ, respectively.

**Table 5.** *Bioconcentration factor* (BCF) values of lettuce, carrot and pepper plants grown in soil treated with VAs.

| Veterinary Antibiotics (mg kg$^{-1}$) | | | Bio-Concentration Factor (µg kg$^{-1}$) | | |
|---|---|---|---|---|---|
| | | | Lettuce | Carrot | Pepper |
| SHOOT | OTC | 10 | $0.069 \pm 0.003$ | $0.010 \pm 0.054$ | $0.033 \pm 0.010$ |
| | | 50 | $0.240 \pm 0.008$ | $0.043 \pm 0.013$ | $0.109 \pm 0.003$ |
| | | 100 | $2.307 \pm 0.008$ | $0.610 \pm 0.009$ | $0.102 \pm 0.003$ |
| | SDZ | 10 | $0.286 \pm 0.002$ | $0.018 \pm 0.023$ | $0.066 \pm 0.005$ |
| | | 50 | $2.373 \pm 0.006$ | $1.069 \pm 0.006$ | $0.727 \pm 0.002$ |
| | | 100 | $6.103 \pm 0.005$ | $3.088 \pm 0.004$ | $1.227 \pm 0.001$ |
| | STR | 10 | - | - | - |
| | | 50 | - | - | - |
| | | 100 | - | - | - |
| ROOT | OTC | 10 | $0.065 \pm 0.013$ | $0.515 \pm 0.009$ | $0.134 \pm 0.091$ |
| | | 50 | $0.177 \pm 0.004$ | $1.517 \pm 0.005$ | $0.374 \pm 0.094$ |
| | | 100 | $0.243 \pm 0.003$ | $2.132 \pm 0.003$ | $0.474 \pm 0.118$ |
| | SDZ | 10 | $0.227 \pm 0.004$ | $1.250 \pm 0.003$ | $0.513 \pm 0.016$ |
| | | 50 | $1.485 \pm 0.021$ | $5.003 \pm 0.003$ | $1.016 \pm 0.057$ |
| | | 100 | $4.671 \pm 0.041$ | $8.859 \pm 0.002$ | $3.361 \pm 0.099$ |
| | STR | 10 | - | - | - |
| | | 50 | - | - | - |
| | | 100 | - | - | - |

± represents standard error. SDZ = Sulfadimethoxine; OTC = Oxytetracycline; STR = Streptomycin.

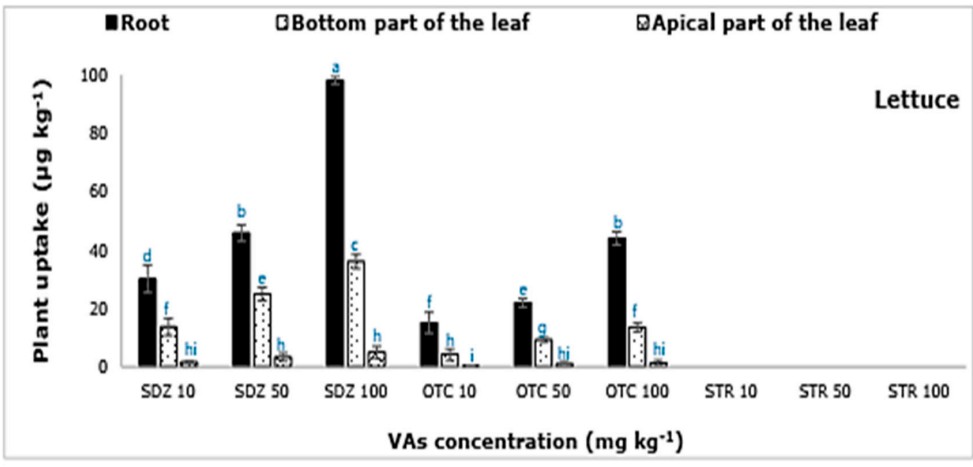

**Figure 1.** *Cont.*

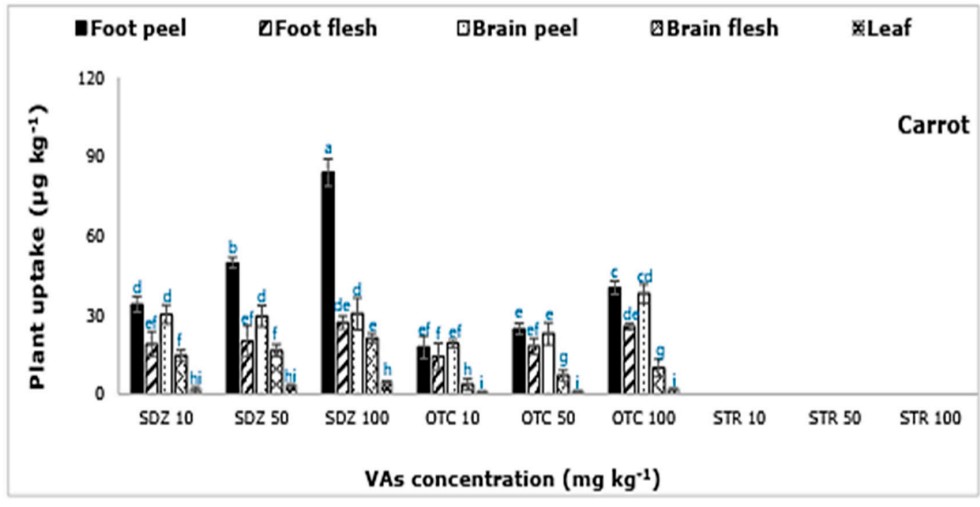

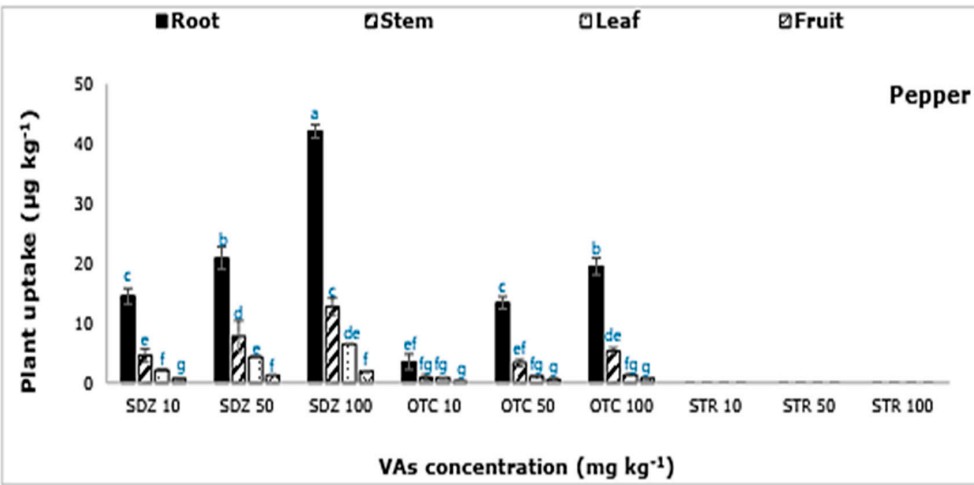

**Figure 1.** Uptake of veterinary antibiotics (VAs) by different parts of lettuce, carrot, and pepper grown in soil treated with VAs. Error bars represent standard error. Concentration values followed by different letters are significantly different ($p < 0.05$) according to Duncan's multiple range test. SDZ = Sulfadimethoxine; OTC = Oxytetracycline; STR = Streptomycin.

The level of VAs sensitivity and eventual bio-accumulation was greatly affected by plant type. For instance, carrot is a root vegetable and significantly affected by VAs. Since roots are the first plant organ to come in contact with soil and its contaminants, carrot a root vegetable was more sensitive to VAs. As a result, there was a higher accumulation of VAs by carrot roots that functions as its primary storage organ. A study funded by the United States Department of Agriculture showed that root crops such as carrots and radishes, that directly come in direct contact with soil, may be particularly vulnerable to VAs contamination [27]. Some of the staples enjoyed worldwide are root vegetables, and the uncontrolled introduction of VAs in the agriculture ecosystem can jeopardize food security. Alternately, lettuce plants accumulated VAs mostly in the shoot (leaves), and the BCF values for VAs in the shoot were in the order: lettuce > pepper > carrot. Lettuce is a leafy vegetable; therefore, VAs were accumulated mostly in the leaf, its primary storage organ. Among the three test plants, the fruit plant (pepper) had the least accumulation of VAs. This study reported a species variation in VA absorption and accumulation, depending upon the plant type and the nature of the plant storage system.

### 3.3. Detoxification of VAs

In this study, we failed to detect the metabolites of the active molecules SDZ and OTC in the plant samples. Some possible reasons for this can be-

- Metabolites not formed during the experimental period.
- Metabolites formed are processed by the plant detoxification system and were beyond the detection limit (at the time of analysis).

The phase I (CPR) and phase II (GST) detoxifying enzymes, of the plant detoxification system, were analyzed. The activity of the CPR enzyme increased with increasing concentration of SDZ and OTC antibiotics in the soil-manure mixture, irrespective of the plant type (Figure 2). On the other hand, there was no significant difference between control and STR antibiotic. CPR enzyme activity ranged from 3.69 to 9.17, 1.69 to 4.70, and 6.98 to 10.47 mmol min$^{-1}$ for lettuce, carrot, and pepper, respectively. CPR enzymes function to metabolize potentially toxic compounds, including drugs. Therefore, increased CPR enzyme activity signifies the onset of detoxification mechanism or the increase in metabolic activity. A previous study has reported the induction of CPR activity in Oryza sativa exposed to the herbicide atrazine [27]. The five-liganded heme-proteins of cytochromes P450 can exist as many isoforms, which enables them to accommodate a wide range of substrates and catalyze a variety of reactions [28]. However, we were unable to find any published reports of plant CPR activity under VA stress. The highest CPR activity of 10.47 mmol min$^{-1}$ reported at OTC 50 mg kg$^{-1}$ in pepper and the lowest (1.69 mmol min$^{-1}$) at SDZ 10 mg kg$^{-1}$ in carrot. The response of the CPR varied depending on the plant species, the plant tissue analyzed, and the intensity of stress (VAs concentration).

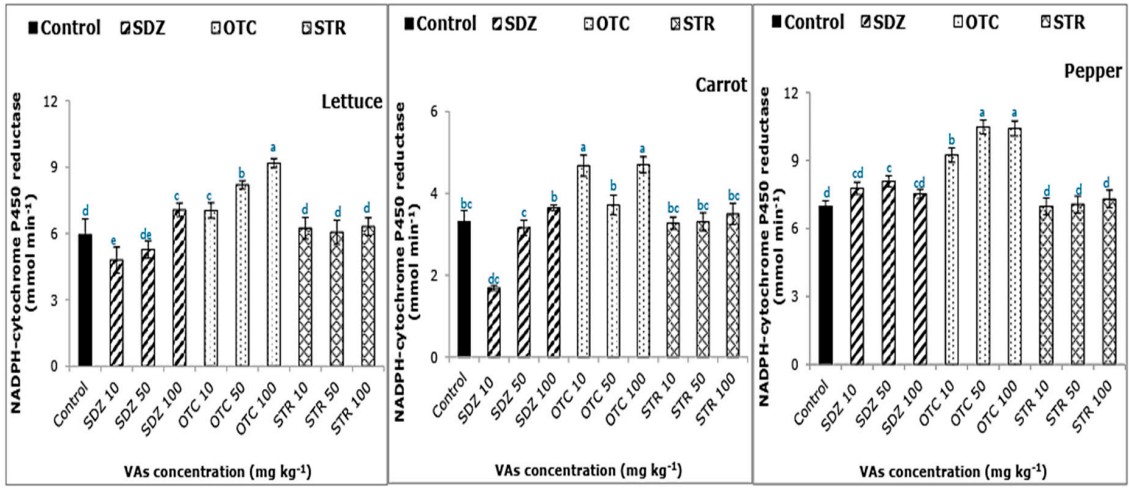

**Figure 2.** NADPH-cytochrome P450 reductase activity of lettuce, carrot, and pepper grown in soil treated with VAs. Error bars represent standard error. Concentration values followed by different letters are significantly different (*p* < 0.05) according to Duncan's multiple range test. SDZ = Sulfadimethoxine; OTC = Oxytetracycline; STR = Streptomycin.

With increasing concentrations of SDZ and OTC in the soil-manure mixture, a gradual increase in GST activity was observed (Figure 3). A previous study showed a significant increase in GST activity of pinto beans grown un chlortetracycline treated soil [11]. GSTs are known to safeguard the cells against chemical-induced toxicity and provide tolerance [29]. In plants, GSTs reportedly provide tolerance to herbicides [30]. In carrot, the overall GST activity for SDZ and OTC was lower, in comparison with control, enforcing the inhibitory effect of SDZ and OTC on GST activity. Alternately in lettuce, low doses (10 mg kg$^{-1}$) resulted in significant inhibition of GST activity. On the contrary, in comparison with control, no significant difference (*p* < 0.05) in GST activity was observed with STR treatment.

Therefore, under VA stress, there was an onset of the plant detoxification system, which varied depending upon the type of VAs and plant species.

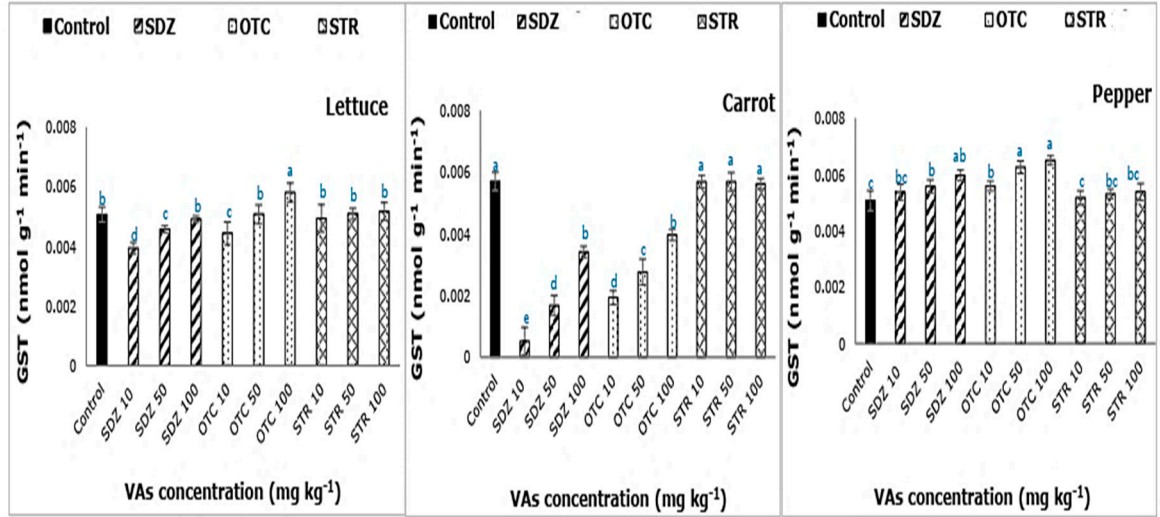

**Figure 3.** *Glutathione-s-transferase* (GST) activity of lettuce, carrot and pepper grown in soil treated with VAs. Error bars represent standard error. Concentration values followed by different letters are significantly different ($p < 0.05$) according to Duncan's multiple range test. SDZ = Sulfadimethoxine; OTC = Oxytetracycline; STR = Streptomycin.

### 3.4. Secondary Experiments

In Figure 4, the mycorrhizal frequency (F%) of lettuce, carrot, and pepper plants grown in VAs amended soil is given. Upon comparison with control, F% was affected the most by SDZ, followed by OTC antibiotics. The effect of STR antibiotics on F% of lettuce, carrot, and pepper plants was found not to be significantly different ($p < 0.05$) from control. At minimum VAs dosage of 10 mg kg$^{-1}$, the frequency of mycorrhizal colonization was higher than that of control. As described in previous studies, under abiotic stress conditions, increased colonization of plant roots by AM fungi is observed [31–33]. An increase in F% at low concentrations followed a steady decline with the increasing dosage of VAs in the soil-manure mix. The formation of mycorrhizal structures in root is associated with the effective colonization of root by various AMF species [31,33]. Roots being the first plant organ to come in contact with the VAs, the physiology of the root affected by VAs treatment, thereby affected the intensity of mycorrhizal colonization. The observed F% varied depending on plant type indicating species variability. The F% of carrot, a root vegetable, reporting the highest accumulation of VAs was affected the most, followed by lettuce and then pepper.

Figure 5 depicts the free proline content in the root and leaf tissues examined. In all three plants, we observed a variation in the accumulation of free proline in response to VAs stress. Free proline concentration increased with increasing abiotic stress (VAs concentration) and was the highest in carrot plants followed by lettuce and then pepper plants. The highest proline accumulation occurred at the highest VAs concentration of 100 mg kg$^{-1}$. As reported in a previous study, the accumulation of proline in *Medicago polymorpha* and *Medicago ciliaris* under NaCl stress, was the greatest at the highest NaCl concentration of 100 mM [34]. Proline contributes to osmotic adjustment, stabilization, and protection of membranes integrity and macromolecules from the damage as a ROS scavenger [35,36]. It is significant in maintaining growth when plants are under stress [37]. Under unfavorable conditions, plants accumulate amino acid metabolites, a constituent of proteins, that play a significant role in plant development and metabolism. Among the amino acids, 'proline' has been reported to play a beneficial role in plants exposed to various stress conditions [38]. Proline plays three significant roles during stress, i.e., as a metal chelator, antioxidant molecule, and a signaling molecule. The high proline concentration in roots, in comparison to leaves, is likely due to the higher concentration of VAs in the roots. Previous studies have also reported an increase in proline accumulation in plants under stress [39].

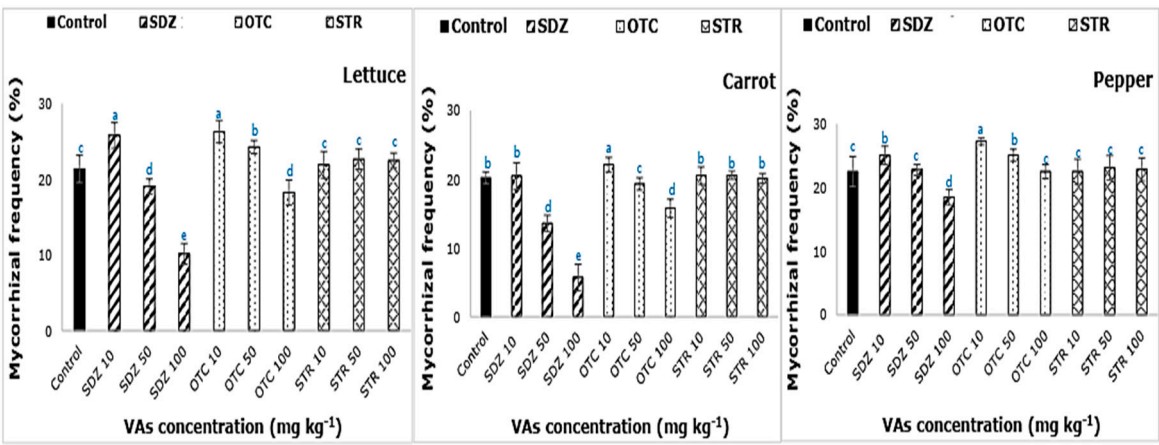

**Figure 4.** Mycorrhizal frequency of lettuce, carrot, and pepper grown in soil treated with VAs. Error bars represent standard error. Concentration values followed by different letters are significantly different ($p < 0.05$) according to Duncan's multiple range test. SDZ = Sulfadimethoxine; OTC = Oxytetracycline; STR = Streptomycin.

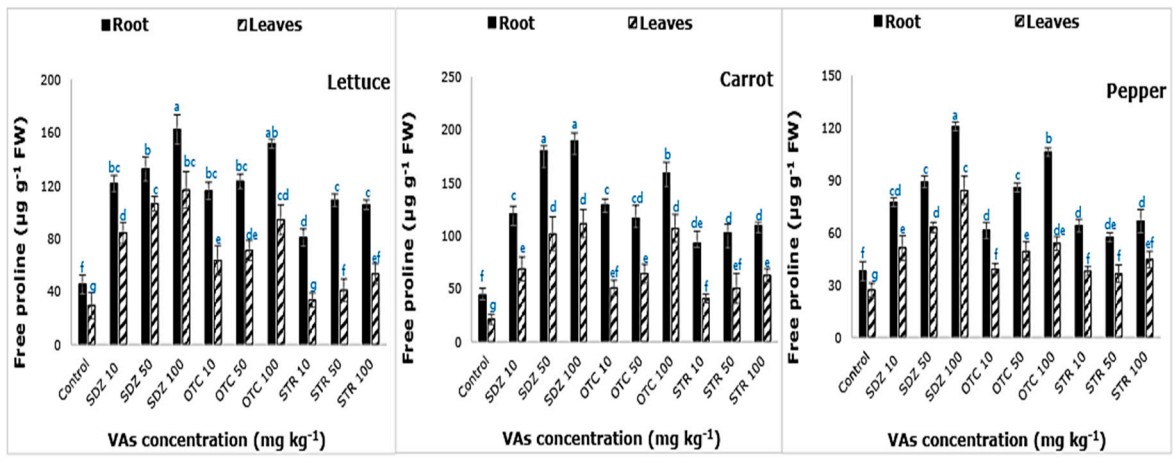

**Figure 5.** Free proline content of lettuce, carrot, and pepper grown in soil treated with VAs. Error bars represent standard error. Concentration values followed by different letters are significantly different ($p < 0.05$) according to Duncan's multiple range test. SDZ = Sulfadimethoxine; OTC = Oxytetracycline; STR = Streptomycin.

Free proline content varied between the plant roots and leaves. Free proline content was higher in the root than in leaves, likely due to the higher accumulation of VAs in plant root. Another possibility is the direct exposure of the roots to the soil contaminated with VAs. The highest accumulation of free proline occurred in carrot roots at 100 mg kg$^{-1}$ SDZ ($189.84 \pm 7.23$ µg g$^{-1}$).

### 3.5. Environmental Risk Assessment

Risk quotients and the corresponding levels of potential ecological risks of the three VAs in the test plants, in sandy clay loam soil, are given in Table 6. When RQ equals or exceeds 1, an ecological "high risk" is suspected. "Low risk" is suspected when $0.01 < RQ < 0.1$, whereas, $0.1 < RQ < 1$ indicates "medium risk". Based on the result, SDZ antibiotics posed a high acute ecological risk to the test plants. On the other hand, OTC and STR antibiotics posed a medium and low acute ecological risk, respectively.

**Table 6.** Estimated risk quotients and corresponding levels of potential acute ecological risks.

| VA's | Plant | PNEC | PEC (mg kg$^{-1}$) | RQ | ERA |
|------|-------|------|---------------------|-----|-----|
| SDZ | Lettuce | $2.107 \times 10^{-3}$ | $3.684 \times 10^{-3}$ | 1.785 | High |
| | Carrot | $1.175 \times 10^{-3}$ | $3.684 \times 10^{-3}$ | 3.135 | High |
| | Pepper | $3.395 \times 10^{-3}$ | $3.684 \times 10^{-3}$ | 1.085 | High |
| OTC | Lettuce | $1.166 \times 10^{-3}$ | $1.052 \times 10^{-3}$ | 0.902 | Medium |
| | Carrot | $1.411 \times 10^{-3}$ | $1.052 \times 10^{-3}$ | 0.745 | Medium |
| | Pepper | $2.511 \times 10^{-3}$ | $1.052 \times 10^{-3}$ | 0.419 | Medium |
| STR | Lettuce | $6.09 \times 10^{-3}$ | $1.05 \times 10^{-1}$ | 0.017 | Low |
| | Carrot | $3.365 \times 10^{-3}$ | $1.05 \times 10^{-1}$ | 0.031 | Low |
| | Pepper | $1.048 \times 10^{-2}$ | $1.05 \times 10^{-1}$ | 0.1001 | Low |

PNEC data calculated from germination index of lettuce, carrot, and pepper seeds in soil.

## 4. Conclusions

The global consumption of antibiotics increased steadily over the past decade, with a predicted increase of 67% by 2030. Moreover, because of poor regulation, VAs are increasingly found in both natural and human-made environments. The persistent VAs and their metabolites may cause an increase in antibiotic-resistant genes, bio-accumulation, and an overall disruption of ecological balance. In this study, the toxic effects of SDZ, OTC, and STR on lettuce, carrot, and pepper were investigated through: (1) seed germination and growth test in soil, (2) bioconcentration study, (3) mycorrhizal frequency and proline content, (4) activity of detoxifying enzymes, (5) environmental risk assessment. Following VAs introduction, the test plants had differential responses depending upon the type of VAs and the plant. The variability of plant species, with regards to VA sensitivity, was established. SDZ, a sulfonamide, was found to be the most toxic to plant growth in soil.

Similarly, carrot plants were the most sensitive, followed by lettuce and then pepper. A higher VAs accumulation and toxicity in carrot, a root vegetable, was evident from the study. The physiological makeup (molecular weight) and behavior in soil (sorption) were the most likely determinants for the differential toxic effects of the antibiotic compounds. Risk assessment study revealed SDZ, OTC, and STR antibiotics to impose high, low, and medium risk in lettuce, carrot, and pepper plants are grown in VAs amended sandy clay loam soil.

**Author Contributions:** Visualization, resources, supervision, project administration, funding acquisition, writing—review and editing, J.-Y.C. Resources, supervision, project administration, funding acquisition, S.-H.R. Conceptualization, methodology, investigation, formal analysis, software, data curation, writing—original draft preparation, writing—review and editing, R.P.T. All authors have read and agree to the published version of the manuscript.

**Funding:** This project was funded by- Rural Development Administration (PJ013367032019), Government of Korea.

**Acknowledgments:** This project was funded by- Rural Development Administration (PJ013367032019), Government of Korea.

**Conflicts of Interest:** The authors declare no conflict of interest.

## Appendix A

**Table A1.** Properties of veterinary antibiotics used in the study.

| COMPOUND | Sulfadimethoxine | Oxytetracycline | Streptomycin Sulfate |
|---|---|---|---|
| Structure |  |  |  |
| Formula | C12H14N4O4S | C22H24N2O9 | C21H39N7O12 ·1.5 H2SO4 |
| M.W (g mol-1) | 310.33 | 460.439 | 728.69 |
| Family | Sulfonamide | Tetracycline | Aminoglycoside |
| CAS no. | 122-11-2 | 79-57-2 | 3810-74-0 |

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
