# Peer review of "Effect of Sulfadimethoxine, Oxytetracycline, and Streptomycin Antibiotics in Three Types of Crop Plants—Root, Leafy, and Fruit"

_applsci, doi:10.3390/app10031111_

Round 1

Reviewer 1 Report

Dear authors,

In my opinion your work is ready to be pbublish.

Best regards!

Author Response

We sincerely thank you and appreciate your time and effort in reviewing the manuscript.

Reviewer 2 Report

The conclusion / summary could be supplemented with a comment on the environmental risk and risk to human health, caused by the occurrence of these antibiotics in the environment.

Author Response

As per the reviewer's suggestion, we have made additional changes to the conclusion (Lines 378-382). We hope that the changes made are satisfactory.

Reviewer 3 Report

This is an interesting study, dealing with a topic that has not been yet extensively studied. Unfortunately, it is very difficult to read this manuscript because it requires extensive language revision. I started to read it carefully, but than I lost will and I went through it very rapidly. Even if it can be of interest, in the present form it is not acceptable. I sugggest the authors to rewrite the manuscript so that it can be easier to read. In parallel, I invite the authors to specify better the experimentla protocl in particular the comparison with control soil.

Author Response

To address language concerns, we too help from a native speaker (Lydia Lombardi- [email protected]) along with online language correction service (Grammarly Premium)

Following the reviewer's suggestion, we have designated the control and test soils used in the study clearly (Lines 89-92). We hope that the changes made are satisfactory.

This manuscript is a resubmission of an earlier submission. The following is a list of the peer review reports and author responses from that submission.

Round 1

Reviewer 1 Report

The MS applsci-654353 describes the uptake and effect of three veterinary antibiotics (VAs)in three plants. Although the topic of the MS is really interesting, I have serious reservations about the analytical procedures and the obtained results as no internal standards were used during sample extractions. Consequently, the detected amounts of VAs in the soil after experiment (Table 4) were very low (less than one tenth of added amount), the detected amounts of VAs in all plant part represent less than hundredth and none metabolite or degradation product was detected (their absence is hardly believable)… Where is the most of VAs that have been added into the soil??? In addition, the text of MS deserves the substantial improvement of English to be more understandable. I cannot recommend publication of this MS in present form.

Reviewer 2 Report

The manuscript contains useful information. However, the MS is written with simple and broken language, thus, English is the major shortcoming of this study and needs to be corrected. I have had difficulties following and understanding some parts of this study. I provided suggestions for authors to correct these unclear points and can be seen on attached PDF file. I also couldn’t help but asked some questions that I believe answers would provide additional useful information about the study and methods used, and help readers to better understand the results. My major reservation is that some conclusions that was drawn were not matching the data presented. Many details are omitted or haphazardly provided so that it was very difficult to follow what authors were doing or saying. Thus, I encourage authors to revise their manuscript with help of my comments (attached on PDF).

Reviewer 3 Report

Dear Authors,

In my opinion the theme of the article is very actual and interesting for the readers of the journal.

The authors evaluated plant uptake of three different veterinary antibiotics (sulfadimethoxine, oxytetracycline, and streptomycin) in lettuce, carrot and pepper grown in a sandy clay loam soil adulterated with fertilizer with antibiotics.

The results showed that sulfadimethoxine and oxytetracycline were taken up by all three plants, with concentrations in plant tissue ranging from 0.1 to 1.2 mg kg−1 dry weight; the antibiotic concentration in plant tissues increased with corresponding increase of antibiotic in manure.

Sulfadimethoxine was found to be the most toxic to plant growth in soil.

From this study variability of plant species to veterinary antibiotics sensitivity was also established.  Carrot was the most sensitive followed by lettuce and then pepper. The direct contact of carrot to the veterinary antibiotics in soil as it is a root plant resulted in higher accumulation and toxicity. The differential toxic effects of the antibiotic compounds were due to their physiological makeup (molecular weight) and behavior in soil (soil properties, such as sorption).

An increase in enzymes activity with increasing sulfadimethoxine and oxytetracycline concentration was evidenced. However, the activity of plant detoxification enzymes under streptomycin treatment was found to not be significantly different from control.

These results alert for potential human health concerns of consuming low levels of antibiotics from produce grown on adulterer soils.

The paper is well structured, the title and abstract clearly describe the content of the manuscript. However, I recommend that a native speaker should correct carefully through the manuscript.

In my opinion minor revision is needed.

Best regards